# The ITS region provides a reliable DNA barcode for identifying reishi/lingzhi (*Ganoderma*) from herbal supplements

Tess Gunnels[1,2], Matthew Creswell[2], Janis McFerrin[2], Justen B. Whittall[1]*

1 Department of Biology, Santa Clara University, Santa Clara, California, United States of America,
2 Oregon's Wild Harvest, Redmond, Oregon, United States of America

* jwhittall@scu.edu

## Abstract

The dietary supplement industry is rapidly growing yet, a recent study revealed that up to 60% of supplements may have substituted ingredients, some of which can be harmful contaminants or additives. When ingredients cannot be verified morphologically or biochemically, DNA barcoding complemented with a molecular phylogenetic analysis can be a powerful method for species authentication. We employed a molecular phylogenetic analysis for species authentication of the commonly used fungal supplement, reishi (*Ganoderma lingzhi*), by amplifying and sequencing the nuclear ribosomal internal transcribed spacer regions (ITS) with genus-specific primers. PCR of six powdered samples and one dried sample all sold as *G. lucidum* representing independent suppliers produced single, strong amplification products in the expected size-range for *Ganoderma*. Both best-hit BLAST and molecular phylogenetic analyses clearly identified the presence of *G. lingzhi* DNA in all seven herbal supplements. We detected variation in the ITS sequences among our samples, but all herbal supplement samples fall within a large clade of *G. lingzhi* ITS sequences. ITS-based phylogenetic analysis is a successful and cost-effective method for DNA-based species authentication that could be used in the herbal supplement industry for this and other fungal and plant species that are otherwise difficult to identify.

## Introduction

Molecular barcoding is an efficient tool for identifying macroscopic, microscopic and biochemically enigmatic samples [1]. It has been applied across the tree of life [2, 3] and is increasingly employed in identifying the provenance of unidentifiable food products in restaurants [4] and in retail [5–7]. Leveraging the DNA content of processed living organisms that is not otherwise identifiable holds great prospects for quality control—especially helpful for authenticating the ingredients and avoiding contaminants and additives that may cause allergic reactions for consumers [8], although DNA of a species can be present long after losing the biological activity of its compounds (i.e. after an aggressive processing). This is particularly relevant in the herbal supplement industry, where safety and effectiveness are loosely regulated

**Data Availability Statement:** DNA sequences are available from Genbank (accession numbers MT994154, MT994155, MT994156, MT994157, MT994158, MT994159, MT994160, MT994161,

MT994162). All other relevant data are within the manuscript and its Supporting Information files.

**Funding:** Oregon's Wild Harvest, the herbal supplement company, provided funding for supplies, sequencing and hourly wages to an undergraduate researcher (TG). The specific roles of this author is articulated in the 'author contributions' section. The funders had no role in study design, data collection and analysis, decision to publish, or preparation of the manuscript.

**Competing interests:** MC and JM are employees of Oregon's Wild Harvest, an herbal supplement company included in our sampling that sells an astragalus reishi blend. As an undergraduate researcher, TG was partially funded by Oregon's Wild Harvest. This does not alter our adherence to PLOS ONE policies on sharing data and materials. There are no other relevant declarations relating to employment, consultancy, patents, products in development, or marketed products, etc.

by the FDA through the Dietary Supplement Health and Education Act of 1994, which requires the manufacturer to ensure the safety and effectiveness of a supplement [9]. The Federal Food, Drug, and Cosmetic Act requires that manufacturers and distributors who wish to market dietary supplements that contain "new dietary ingredients" (not marketed in a dietary supplements before October 15, 1994) notify the Food and Drug Administration about these ingredients (Section 413(d) of [10]). Under this Act, it is the responsibility of the manufacturer or distributor to assess whether a dietary supplement will be safe to use [10].

The herbal supplement industry is a growing enterprise, expected to amount to $104.78 billion dollars or more by 2025 [11, 12], yet a recent study revealed that up to 60% of herbal supplements have substituted ingredients not listed on their labels, some of which can be harmful contaminants or additives [8]. For both marketing advantage and ethical concerns, suppliers must ensure accurate identification of all ingredients in their products [9]. Moreover, dietary supplement regulations require a manufacturer to perform identity testing on 100% of incoming lots of dietary ingredients, except when it has petitioned the FDA for a special exemption [10, 13]. For some manufactures, accurate identification of species, complete listing of ingredients, and precise reporting of potency are paramount. Furthermore, retailers are expected to exercise due diligence regarding oversight of suppliers. This is especially important since a large portion of the population consuming herbal supplements are doing so because their health is already compromised [14].

Reishi (*Ganoderma* spp. P.Karst.) is one of the oldest herbal medicines in recorded history [15, 16] and estimated to represent 2% of the herbal supplement industry [14]. It is recommended as an anti-inflammatory and to enhance immunity [17, 18]. After being cultivated on rice, most reishi products are ground to a powder and sold in capsules as herbal supplements. Although the glossy, lignicolous, leathery, shelf-like polypore fruiting bodies of this group of laccate *Ganoderma* species are distinctive when fresh, once pulverized along with the rice medium (which often constitutes >50% of the dry weight), the powder is not easily differentiable macroscopically, microscopically, or biochemically [18, 19]. For example, using biochemistry, Wu et al. [19] could only verify 26% of 19 reishi supplements purchased in the United States as true reishi (they use "*G. lucidum*", but we will use *G. lingzhi* Wu, Cao & Dai for true reishi heretofore).

Adding to the difficulty of identifying processed reishi is the taxonomic confusion surrounding the species within *Ganoderma* [20]. The genus consists of approximately 80 species that fall into five or six clades—one of which is centered around *G. lingzhi* (Clade A) and another clade includes the true *G. lucidum* P.Karst. (Clade B) [21–26]. Because of their wood-decaying capabilities, several *Ganoderma* species have been investigated for biopulping [14] and bioremediation [27], however, it is most prized as a "model medicinal mushroom" [28] because of the putative health benefits of the triterpenoids and polysaccharides [29, 30]. These clades include several well-supported phylogenetic lineages that received unstable taxonomic treatments in the past [22, 24, 25]. According to a thorough morphological and molecular investigation of the commonly cited *G. lucidum* and the actual medicinal mushroom, *G. lingzhi* "the most striking characteristics which differentiate *G. lingzhi* from *G. lucidum* are the presence of melanoid bands in the context, a yellow pore surface and thick dissepiments (80–120 μm) at maturity" [23]. *Ganoderma lucidum* (including *G. tsugae* Murrill) can be found in the wild from Europe to northeastern China (some have likely escaped from cultivation in California and Utah [22]), whereas *G. lingzhi* is restricted to Asia [18]. Fresh *G. lingzhi* has higher levels of triterpenoids than *G. lucidum* which may be responsible for the suggested physiological effects of *G. lingzhi* in humans. This biochemical result also supports the distinctiveness and the commercial importance of differentiating these often confused taxa [18, 19]. The history of taxonomic confusion surrounding *G. lucidum* and *G. lingzhi* [18, 26, 31] has been

largely resolved by recent morphological comparisons [23], biochemical investigation [18], and molecular phylogenetic analyses [22–24, 32]. In particular, molecular phylogenetic analyses place G. *lucidum* and its closest relatives (*G. oregonense*, *G. tsugae*, and *G. carnosum*) in Clade B and the medicinally important *G. lingzhi* in Clade A with numerous other closely related species [22, 24].

The nuclear ribosomal internal transcribed spacer region (ITS) is an informative DNA region for barcoding plants and fungi [33–35]. It consists of two hypervariable spacers of approximately 200-250bp flanked by the 18S (small) subunit and 28S (large) subunit rDNA and separated by the 5.8S rDNA [36, 37]. Primers designed to bind to highly conserved portions of the 18S and 28S subunits have been widely used across plants and fungi [37]. However, lineage-specific primers have been developed for many groups of fungi to help diagnose the presence or absence of particular species [23]. Lineage-specific primers can also improve PCR specificity especially when working with compromised DNA templates that may be degraded, contain inhibitors, or be composed of a mixture of species. *Ganoderma*-specific primers developed by Cao et al. [23] have been shown to improve PCR specificity. These primers have been used for barcoding reishi herbal supplements in previous studies (see [38] with limited sampling of reishi samples (n = 4) and [14] with a broader retail sampling (n = 14), but unclear how many unique suppliers were represented in the latter).

Although considerable attention has been given to the identification of the best barcoding loci and the development of unique and creative applications [34], less explicit attention has been paid to the analysis of the data. The two main approaches for analysis of the DNA sequences arising from barcoding investigations are genetic distance-based measures (e.g., best-hit BLAST or nearest neighbor analysis) and phylogenetic methods (e.g., maximum likelihood or Bayesian tree-building algorithms). Some studies rely solely on best-hit BLAST [39] or otherwise crude phylogenetic approaches [40, 41] sometimes without assessment of the uncertainty [8, 42]. Genetic distance-based measures are known to fail in several common situations such as variable rates of molecular evolution [43, 44], gene duplication [44, 45] and changes in a gene's composition [46]. Empirically, genetic distance-based approaches and phylogenetic methods for barcoding analysis are rarely compared explicitly even though they can produce conflicting identifications [47, 48].

Herein, we present an efficient method for unambiguous identification of the herbal supplement, reishi (*G. lingzhi*). We report successful DNA extraction, PCR amplification, and DNA sequencing of the ITS region from store-bought reishi samples. We compare the results from best-hit BLAST with two molecular phylogenetic approaches to determine if the species in the store-bought samples are correctly labeled or not.

## Materials and methods

### Sampling

Store-bought samples were collected from multiple nutritional supplement retailers representing seven distinct suppliers of cultivated fungal products (Table 1). Four samples were encapsulated powders and two were loose powders, all of which purport to contain reishi, or "*Ganoderma lucidum*", based on the product's labeling. Of the seven supplements sampled, four were labeled as containing only mushroom mycelial biomass, two samples claimed to contain both mycelia and fruiting body, and one sample did not specify. The powdered samples varied in color, texture, and smell. All powdered samples were macroscopically unidentifiable as a mushroom and for Powder #1, only mycelia were observed under compound microscope (40-100x) (Cresswell and McFerrin, unpublished data).

**Table 1. Sampling information for powdered and fresh samples that were store-bought or wild-collected.**

| Sample | Species (as advertised) | Sample Information |
|---|---|---|
| Powder #1 | *Ganoderma lucidum* | Store-bought: Oregon's Wild Harvest Astragalus-Reishi |
| | | Supplier: Oregon's Wild Harvest |
| Powder #2 | *Ganoderma lucidum* | Store-bought: Host Defense Reishi |
| | | Supplier: Fungi Perfecti |
| Powder #3 | *Ganoderma lucidum* | Store-bought: Solaray Reishi Mushroom |
| | | Supplier: Nutraceutical Corp. |
| Powder #4 | *Ganoderma lucidum* | Store-bought: The Vitamin Shoppe Reishi Mushroom |
| | | Supplier: Gourmet Mushroom Inc. |
| Powder #5 | *Ganoderma lucidum* | Store-bought: Eclectic Institute Fresh Freeze Dried Reishi Mushrooms |
| | | Supplier: Eclectic Institute Inc. |
| Powder #6 | *Ganoderma lucidum* | Store-bought: Now Rei-Shi Mushrooms |
| | | Supplier: Now Foods |
| Fresh #1 | *Ganoderma brownii* | Wild-collected: De Laveaga County Park, Redwood Loop Trail, approximately 50 m NE of crooked tree picnic area, common among dead *Umbellularia californica*, Santa Cruz, CA, USA (36.999720, -122.000360) |
| Fresh #2 | *Fomitopsis pinicola* | Wild-collected: Pogonip County park, Fern Trail, approximately 0.5 km south of junction with Spring Trail, on dead *Quercus agrifolia*, Santa Cruz, CA, USA (37.001568, -122.042023) |
| Fresh #3 | *Ganoderma lucidum* | Store-bought: Staff of Life Organic Reishi Mushroom |
| | | Supplier: Mycological Natural Products |

A fresh mushroom sample advertised as "organic reishi mushroom" was collected from the bulk herb section of Staff of Life natural goods store, Santa Cruz, CA in July of 2018 and was also evaluated based on its morphological characteristics (Fresh #3 in Table 1). The sample had been cut into strips of approximately 6 x 1 cm from cross sections of the fruiting body. The sample appeared woody in texture with extensive pore-containing regions similar to morphologically identified samples of the complete fruiting body.

Two additional fresh samples were collected from the wild (Santa Cruz County, CA, USA; Table 1) and used as positive controls for DNA extraction, PCR and sequencing (Table 1, Fresh #1 and Fresh #2). Samples were morphologically identified as *Ganoderma brownii* (Murrill)Gilb. and *Fomitopsis pinicola* (Sw.)P.Karst. [49]. These two closely related genera can be distinguished by the presence (*Ganoderma*) or absence (*Fomitopsis*) of bruising on the white pores of the fruiting body's underside [49]. All samples were stored at room temperature until the DNA could be extracted.

## DNA extraction

For each nutritional supplement, two subsamples were taken from each sample and DNA was extracted from each of them. Encapsulated samples were opened and only the powder contained within was used. Field collected samples were dissected and cut into smaller pieces for further morphological evaluation and then prepared for DNA extraction. Fresh tissue was removed from the underside of the fruiting body and cut into 2mm x 5mm rectangles for homogenization. Approximately 30–100 mg of material was homogenized in QIAGEN's DNEasy Plant Mini Kit extraction buffer using a BeadBeater with 4 x 3.2 mm steel beads in XXTuff 2mL O-ring screw cap tubes (Biospec, Bartlesville, OK, USA). Following homogenization, DNA extraction was performed using the Qiagen DNEasy Plant Mini Kit following the manufacturer's protocol (QIAGEN, Valencia, CA, USA). Concentration and purity of

extracted DNA was evaluated using a Nanodrop spectrophotometer (NanoDrop Technologies Inc., Wilmington, DE, USA).

## PCR and sequencing

Several fungal ITS primer pairs were tested for initial success of amplification for both fresh and powdered samples (S1 Table) [50]. Among them, the *Ganoderma*-specific primers (G-ITS-F1 and G-ITS-R2) were selected based on consistently producing strong, single bands [23]. These primers were designed to prevent amplification from plant or other fungal DNA, which is a common problem with herbal supplements since they often include a plant-based growing medium or fungal contamination.

Extracted DNA was used as a template in 25 μL PCR reactions. Each reaction consisted of 2.5 μL of MgCl$_2$ (25 mM), 2.5 μL of Taq Buffer B (Mg-free; 10X) (New England Biolabs, Ipswich, MA, USA), 2.5 μL of dNTPs (2.5 mM of each base), 2.5 μL of each of the aforementioned primers (10 μM), 0.25 μL of Taq polymerase (5U/μL) (New England Biolabs, Ipswich, MA, USA) and 1μL of extracted template DNA. A negative control (Milli-Q water in place of DNA template) was included in each PCR to ensure there was no contamination. Amplification took place under the following thermal cycling conditions: initial denaturation at 92˚C for 2 min followed by 35 cycles of 94˚C for 1 min, 55˚C for 45 s, 72˚C for 45 s and a final extension step at 72˚C for 5 min. The PCR products were run on a 1% agarose gel stained with ethidium bromide alongside a 100 bp ladder (New England Biolabs, Ipswitch, MA, USA).

PCR reactions producing single, strong bands, were cleaned-up using shrimp alkaline phosphatase and directly sequenced in both directions using the Applied Biosystems 3730xl DNA Analyzer with the same primers used in PCR (Applied Biosystems, Waltham, Massachusetts, USA). Direct sequencing followed by BigDye Terminator or BigDye Primer methodologies per manufacturer recommendations (Sequetech, Mountain View, CA, USA). Forward and reverse chromatograms for each sample were trimmed to remove the opposing primer sequence and low-quality sequence at the beginning and end of each read and edited to correct any ambiguous base calls. Reads were then aligned to form a single contiguous sequence using the pairwise alignment tool in Geneious Prime (Geneious Prime 2019.0.4, Biomatters, Auckland, NZ).

## Data analysis

**BLAST.** We used Basic Local Alignment Search Tool (BLAST) as the first method of identification for each sample. We used the megablast algorithm to search the nucleotide (nr/nt) collection to find the closest match to our sequences (BLASTDBv4) [51–53]. We compared the BLAST results from full-length sequence queries to the BLAST results of sequences trimmed to the portion of the alignment with maximum overlap with the reference database that were used in our phylogenetic approach (see below).

**Multiple sequence alignment.** We assembled an alignment of related sequences from Genbank (S2 Table). We started by including our top BLAST hits from the full sequence search query described above. If multiple Genbank accessions had equal coverage and identity as the top hit, we took at least one representative of each species which appeared. We also added all the unique Reishi samples (*G. lingzhi* and *G. lucidum*) of ITS using text searches in ENTREZ. Finally, we included representatives of as many *Ganoderma* species we could find using a filtered discontiguous megablast allowing us to limit ourselves to the most highly similar *Ganoderma* accessions, and increase our taxonomic coverage with a diverse and comprehensive reference set for nucleotide alignment and subsequent phylogenetic analysis. Several outgroup

sequences were chosen which included other mushroom species belonging to the same order, *Polyporales*.

Sequences were aligned using the Geneious alignment tool (Biomatters, Auckland, New Zealand). All sequences were trimmed to approximately the same size producing an alignment of consistent length across the available ITS sequences of the Genbank reference set. Sequences with 100% nucleotide match to another sequence of the same species were removed so that only one representative sequence remained to simplify later phylogenetic analyses (S3 Table). If a sequence had a 100% match to a sequence belonging to a different species, both sequences were kept in the alignment to represent the additional taxonomic diversity. After all sequences were trimmed to approximately the same length we repeated the BLAST analysis of each sample to determine if sequence length affected the identity of the unknown samples.

**Phylogenetic analyses.** Maximum likelihood analysis was performed using the RAxML plug-in for Geneious (RAxML 8.2.11) [54]. We applied the GTR + CAT + I model of evolution and employed a rapid bootstrapping algorithm using 1,000 bootstrap replicates. Additionally, the MrBayes 3.2 plugin was used to build a Bayesian phylogenetic inference using Markov chain Monte Carlo (MCMC) algorithm (MrBayes 3.2.6) [55]. The GTR substitution model was used with a proportion invariable, remaining gamma rate variation model. Bayesian analysis ran for 2,000,000 generations. After removing the first 1,000,000 generations as burn-in, we sampled trees every 1000 generations creating a posterior distribution of 1000 trees. The intention of our study is not to disentangle the taxonomic uncertainty regarding *G. lucidum* sensu lato and *G. lingzhi*. Therefore, throughout the results and discussion we have chosen to report the scientific names as they are reported in Genbank although some of these have been suggested to be mislabeled (see the S2 Table in [56]).

## Results

### DNA extraction

The average concentration of DNA in the nine samples was 34.1 ng/uL (range 3.9 to 175.2; S4 Table). The average purity of the DNA measured as the 260/280 ratio was 1.34 (range 0.66 to 1.91; S4 Table).

### PCR and sequencing

To assess successful amplification of the ITS region from newly extracted fungal DNA, PCR with three different primer pairs was performed and samples were visualized with gel electrophoresis. All primer pairs produced visible bands of expected size for the ITS region for both fresh and powdered samples. The PCR products using *Ganoderma*-specific primers were chosen for sequencing and all subsequent analyses based on their increased band intensity compared to other primers. After trimming these newly created sequences for seven samples, lengths ranged from 780 base pairs to 895 base pairs with an average of 854 base pairs. Quality scores (HQ%) for full contiguous sequences of the forward and reverse directions ranged from 75.4% to 96.5% and averaged 91.2%.

### Data analysis

**BLAST.** Our first approach for sample identification was to query Genbank for the top BLAST hit using the full length ITS sequence (Table 2). Of the seven store-bought samples, all yielded a top BLAST result that matched their labeled genus and species ("*G. lucidum*", likely a mislabeled *G. lingzhi*, see Phylogenetic Analyses section below). Top BLAST hits changed to *G.*

**Table 2. BLAST results using full length ITS sequences compared to ITS sequences trimmed to the GenBank reference panel alignment (618 bp) used in phylogenetic analysis.**

| Sample Name | Presumed Species[1] | Genbank Accession | Full Length | | | | Trimmed to Alignment Length | | |
|---|---|---|---|---|---|---|---|---|---|
| | | | Sequence Length (bp) | Top BLAST Hit[2] | GenBank Query Coverage | GenBank Percent Similarity | Top BLAST Hit[2] | GenBank Query Coverage | GenBank Percent Similarity |
| Powder #1 | *G. lucidum* | MT994154 | 824 | *G. lucidum* (MF476200.1) | 100% | 100% | *G. lingzhi* (MH160076.1) | 100% | 100% |
| Powder #2 | *G. lucidum* | MT994155 | 739 | *G. lucidum* (MF476201.1) | 99% | 99% | *G. lingzhi* (MH160076.1) | 99% | 99% |
| Powder #3 | *G. lucidum* | MT994156 | 868 | *G. lucidum* (MF476200.1) | 100% | 100% | *G. lingzhi* (MH160076.1) | 100% | 100% |
| Powder #4 | *G. lucidum* | MT994157 | 868 | *G. lucidum* (MF476200.1) | 100% | 100% | *G. lingzhi* (MH160076.1) | 100% | 100% |
| Powder #5 | *G. lucidum* | MT994158 | 865 | *G. lucidum* (MF476200.1) | 100% | 100% | *G. lingzhi* (MH160076.1) | 100% | 100% |
| Powder #6 | *G. lucidum* | MT994159 | 844 | *G. lucidum* (MF476200.1) | 100% | 99% | *G. lingzhi* (MH160076.1) | 100% | 100% |
| Fresh #1 | *G. brownii* | MT994160 | 848 | *G. australe* (MK968731.1) | 100% | 97% | *G. brownii* (MG279159.1) | 100% | 100% |
| Fresh #2 | *Fomitopsis pinicola* | MT994161 | 780 | *F. pinicola* (EF530947.1) | 100% | 99% | *F. pinicola* (EF530947.1) | 100% | 99% |
| Fresh #3 | *G. lucidum* | MT994162 | 868 | *G. lucidum* (MF476200.1) | 100% | 99% | *G. lingzhi* (MH160076.1) | 100% | 100% |

[1]Presumed species is based on product label for store-bought samples and morphological identification [49] for wild-collected samples.
[2]All top BLAST hits had an E-value of 0.0.

*lingzhi* for all fresh samples when using the trimmed sequences from the 618 bp alignment as described in more detail below (Table 2).

**Multiple sequence alignment.** To further assess the identity of our store-bought and field-collected samples, they were aligned with a reference panel (S2 Table). After trimming the alignment to the length of the shortest sequence in the reference panel and temporarily removing identical sequences (S3 Table), we created a final alignment of 93 sequences measuring 618 base pairs long with 52.2% identical sites (including outgroups). Among these unique sequences, the average pairwise percent identity is 87.6%. Within the *G. lingzhi* clade, there were 91 variable sites (mean pairwise identity = 99.3%). The average genetic identity between our store-bought samples and the most similar Genbank accession was 99.8% (range 99.5–100%).

**Phylogenetic analyses.** The maximum likelihood analysis yielded a moderately resolved tree. Of the 91 distinct branches in the maximum likelihood tree, 31 branches (34%) had bootstrap values greater than 70%, a commonly used cut-off for 95% reliability (Fig 1) [57]. There is a moderately supported *G. lingzhi* clade containing nearly all of the samples labeled *G. lingzhi*, several *G. lucidum* samples, one likely misidentified *G. sichuanense* sample, and all seven of the store-bought herbal supplement samples (bootstrap = 88%) (Fig 1). We also reconstructed a strongly supported clade containing the real *G. lucidum*, *G. tsugae*, *G. oregonense* and *G. carnosum* (Clade B; 100% bootstrap) (Fig 1). We have applied clade names A and B from Loyd et al. [22] and Zhou et al. [24]. Clade A containing *G. tuberosum* and *G. multipileum* appears paraphyletic in Fig 1, however the deepest nodes are only weakly supported (<20% bootstrap) and therefore, not in conflict with previous studies [22, 24].

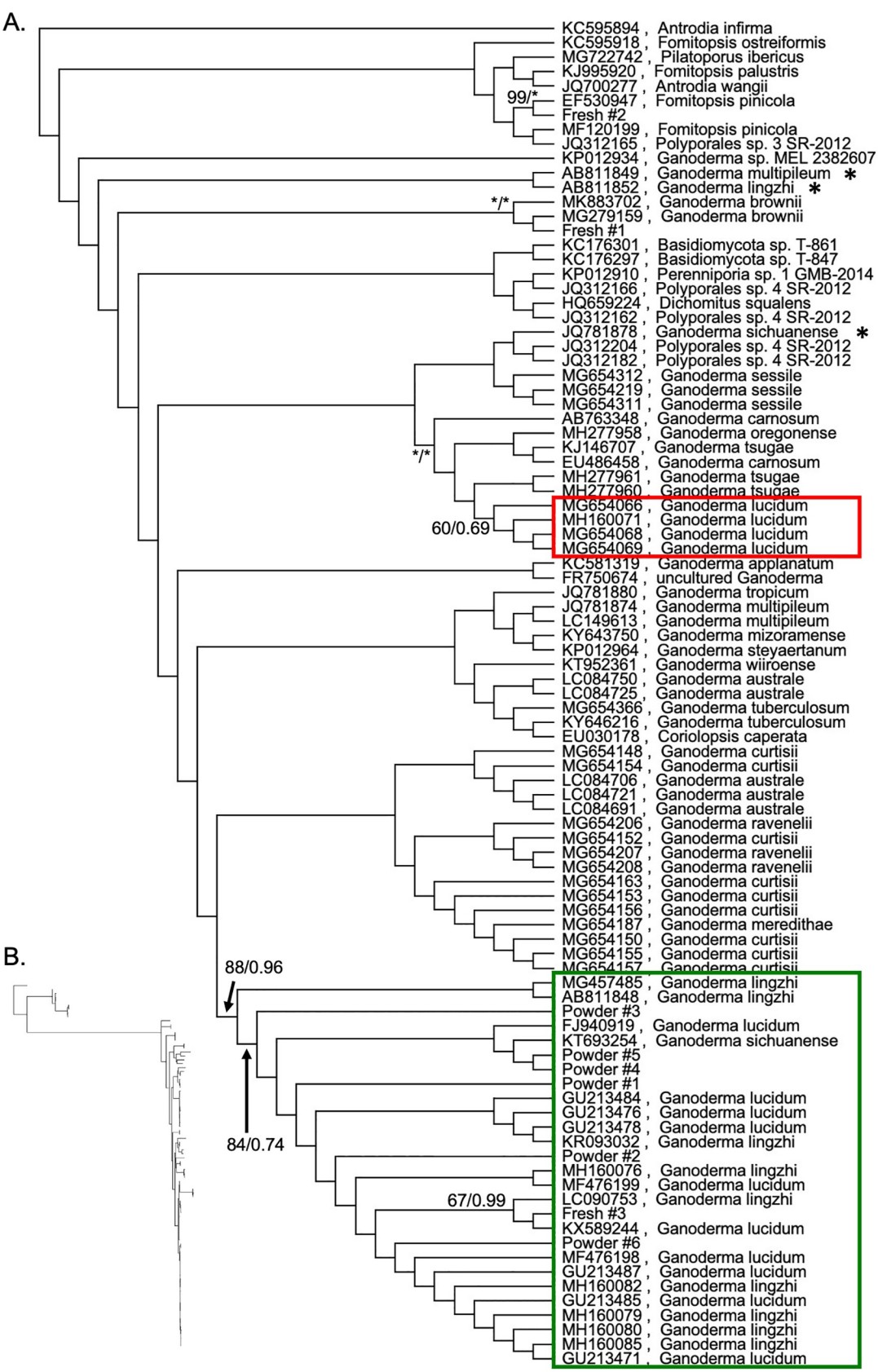

**Fig 1. Maximum likelihood phylogenetic analysis.** RAxML phylogeny including store-bought samples, wild collected samples and the Genbank reference set. (A) Cladogram with branch support at critical nodes indicated along the branches as maximum likelihood bootstrap percentage/ Bayesian posterior probability (asterisks indicate 100% bootstrap and 1.0 posterior probability). Clade names A and B are from Loyd et al. [22] and Zhou et al. [24]. The red rectangle identifies the true *G. lucidum* samples per Loyd et al. [14] and the green rectangle contains the samples referred to as the *G. lingzhi* clade (many *G. lucidum* sequences are misidentified *G. lingzhi*). (B) Maximum likelihood phylogram with unlabeled tips in the same order depicting branchlengths proportional to substitutions per site.

Although there is variation among the reishi samples, there is very little resolution within the *G. lingzhi* clade (a portion of Clade A; Fig 1). The mean pairwise genetic identity among our herbal supplement samples was 99.8%, yet there are only two branches with bootstrap values greater than 70%. The phylogenetic affinities of Powders #1–5 were completely unsupported (Fig 1). Only Fresh #3 had a weak to moderate affinity to "*G. lucidum*" (KX589244) with low bootstrap support (67%). Finally, Powder #6 appears sister to a clade of eight poorly resolved accessions named *G. lingzhi* and *G. lucidum* with very weak bootstrap support (54%; Fig 1).

As a control, we included two fresh samples of wild-collected polypores. Fresh #1 was morphologically identified as *Ganoderma brownii* and was 100% identical to two other *G. brownii* samples (MK883702 & MG279159). Fresh #2 was morphologically identified as *Fomitopsis pinicola* and only had one nucleotide difference (99.8% identical) when compared to the *F. pinicola* Genbank accession that it paired with (EF530947; Fig 1).

The maximum likelihood tree revealed two putatively incorrectly named Genbank accessions worth noting (Fig 1): (1) a *G. lingzhi* sample (AB811852) that is strongly supported as sister to *G. multipileum* (AB811849; 100% bootstrap), but both are very divergent from their Clade A conspecifics; (2) a *G. sichuanense* sample (KT693254) that is nested within the well supported *G. lingzhi* clade (88% bootstrap), yet deeply separated from another sample of the same species (JQ781878) (see [22] for discussion about this taxon).

The Bayesian phylogenetic analysis is largely consistent with the maximum likelihood tree, yet considerably less resolved. Twenty-four branches (26%) have posterior probabilities greater than 0.95 (S1 Fig). Within Clade A, there is a strongly supported subclade containing nearly all of the Genbank accessions named *G. lingzhi*, many erroneously named *G. lucidum* and all of the store-bought samples (posterior probability = 0.96; S1 Fig). Clade B is strongly supported as monophyletic (posterior probability = 1.0) containing a monophyletic lineage of correctly identified *G. lucidum* accessions (per [22]). The two putatively misidentified accessions described for the maximum likelihood analysis above had similarly unexpected phylogenetic affinities in the Bayesian analysis (S1 Fig).

In the Bayesian phylogenetic tree, six of the seven store-bought samples are part of a large unresolved polytomy of accessions named *G. lucidum* and *G. lingzhi* (the true *G. lingzhi* clade). The exceptional sample (Fresh #3) falls within a strongly supported subclade (posterior probability = 1.0) that is composed of an unresolved trichotomy with two other Genbank samples—one labeled *G. lucidum* and one labeled *G. lingzhi* (KX589244 and LC090753, respectively; S1 Fig).

For the Bayesian analysis, the two control samples allied with similar Genbank accessions as in the maximum likelihood analysis. Fresh #1 (morphologically identified as *G. brownii*) allies with the other two *G. brownii* samples with a posterior probability of 1.0 (S1 Fig). Fresh #2 (morphologically identified as *Fomitopsis pinicola)* is strongly supported as sister with a Genbank *F. pinicola* sample (EF530947; posterior probability = 1.0; S1 Fig).

## Discussion

Our study demonstrates that the ITS region provides an efficient barcode for store-bought reishi herbal supplements thereby supporting the conclusions reached by earlier authors

including Loyd et al. [14] and Raja et al. [38]. Amplifiable genomic DNA was successfully extracted from both powdered and fresh samples—all of which closely allied with established *G. lingzhi* samples within Clade A (even though many of those samples and the herbal supplement samples were sold as "*G. lucidum*"). Loyd et al. [14] found widespread label confusion in both "grow your own" kits (15/17) and manufactured herbal supplements (13/14) that were sold as "*G. lucidum*". They used both ITS and tef1-alpha sequences to identify the manufactured supplements were all *G. lingzhi*, except one *G. applanatum* [14]. The label confusion surrounding *G. lucidum* and *G. lingzhi* was likely unintentional due to the taxonomic uncertainty, although there are clear biochemical differences (and therefore potential human physiological consequences) that differentiate the two taxa [18, 19]. In fact, Wu et al. [19] considered 26% of their 19 samples "verified" even though the labels read "*G. lucidum*" and not the correct species name, "*G. lingzhi*." Rampant misidentification of true reishi is highlighted in the authoritative Herbs of Commerce [58] which indicates that the most important species commercially sold under the common name "reishi" are "*G. japonicum*, *G. lucidum*, and *G. tsugae*"–completely neglecting what is now considered true reishi, "*G. lingzhi*" [14, 22, 24, 32].

In general, herbal supplements are notoriously mislabeled—Newmaster et al.'s [8] study of plant herbal supplements found 59% (30 out of 44) had species substitutions and about 33% of these products had fillers or contaminants that were not listed on the product label—some of which could pose health risks to consumers. Herbal Commerce DNA barcoding will continue to be a valuable tool for manufacturers, retailers and consumer-watch groups, especially for herbal supplements like reishi where a lack of morphological and chemical distinctiveness once in powder form is compounded by underlying taxonomic confusion.

All of the samples we examined had BLAST and phylogenetic results suggesting they were clearly members of Clade A sensu Zhou et al. [24] and Loyd et al. [14]. None of our nine distinct distributors sampled contained material belonging to Clade B (*G. lucidum*). Technically all of our samples are misidentified since they are being sold as "*G. lucidum*", yet are molecularly allied with the true reishi samples in Clade A ("*G. lingzhi*"). A similar case of mistaken identity is reported by Loyd et al. [14]. We assume the mislabeling was unintentional and arose from the history of taxonomic confusion surrounding *G. lucidum* vs. *G. lingzhi* (yet recently and lucidly clarified by [32]). This level of mistaken identity (100%) is relatively rare among herbal supplement barcoding studies in general [8]. Although we only included seven store-bought samples, these represent seven distinct suppliers thereby broadening the implications of our study to all the retailers using those suppliers as well, something previous *Ganoderma* retail barcoding studies have not reported (using different retail samples from the same supplier could be considered pseudoreplication; see [14, 38]). Misidentifications can arise at any of the multitude of links that connect the growers with the retailers. Our targeted sampling at the supplier stage clearly indicates that the misidentifications are likely applied early in the process and inherited by the retailers.

Our study does not attempt to resolve the taxonomic ambiguity among closely related species within the genus *Ganoderma* which permeates the available sequence data in Genbank [23, 32, 56]. However, we do not wish to contribute to the confusion so will attempt to reconcile some of the Genbank names with the recent literature in regard to reishi here. Clade A includes a strongly supported lineage of the medicinally important reishi (also known by the common name "lingzhi") which is properly named *G. lingzhi* and restricted to Asia (see [32] for nomenclatural justification; also see Correction in [56]). These samples should all be identified as *G. lingzhi* according to Zhou et al. [24], Patterson & Lima [32], and Loyd et al. [22]. Alternatively, we have recovered a clade of four genetically distinct *G. lucidum* sequences (bootstrap = 60%) which ally with three other taxa to comprise the very strongly supported Clade B (100%). The real *G. lucidum* is native to Europe, closely related to North American *G.*

*oregonense* and *G. tsugae*, and most likely introduced to Utah and California, USA according to Loyd et al. [22]. In comparison to Cao et al. [23] who examined four nuclear genes including ITS (yet only four samples of *G. lingzhi* and *G. lucidum* and a total of 13 species), our analysis has 10 *G. lingzhi* and 11 *G. lucidum* samples) and more species overall (n = 29), yet limited to the single barcoding locus, the ITS region.

More broadly, our phylogenetic results (Fig 1) are generally congruent with previous studies employing the ITS region [22–24, 56]. They all report similar clades that we have identified in our results, yet they often report higher confidence likely due to the inclusion of more loci. Although we have chosen to report the Genbank organism fields as they are in the database, we highlight the taxonomic confusion around these lineages and anticipate their realignment in Genbank in the near future.

ITS variation within the *G. lingzhi* clade allowed us to further partition our store-bought samples. Most samples were part of a large unresolved polytomy, but in two cases, there were distinct phylogenetic affinities suggesting separate sources. The intraspecific variation in ITS could prove valuable for tracing the intraspecific provenance of some reishi herbal supplements, but will likely need to be complemented with additional rapidly evolving loci (e.g., tef1-alpha, see [14, 22, 24]).

Methodologically, BLAST and phylogenetic analyses agreed on the provenance of all of the store-bought samples. When the rates of molecular evolution are relatively constant among the samples, in the absence of gene duplication, and when gene structure is conserved (such as for the ITS region), BLAST and phylogenetic methods are predicted to converge on similar identifications [43, 44, 46]. However, when any of those characteristics are violated, genetic distance-based approaches, such as BLAST, that rely on a local alignment algorithm (some modification of [59]) can be misleading. Alternatively, phylogenetic analysis relies on a global alignment algorithm [60] spanning the entire length of the locus being compared and is more likely to identify the evolutionary history of the samples for that locus [44], yet is most rigorously employed with a model-based approach (e.g. maximum likelihood and Bayesian methodologies) compared to a distance-based approach that is commonly found in the barcoding literature [61]. UNITE is a noteworthy database and search tool for identifying fungal ITS sequences to species using some objective sequence-based cutoffs that should be considered in future barcoding studies [62]. Our results were generally robust to whether we used the entire ITS region or the trimmed region of overlap used in the multiple sequence alignment and subsequent phylogenetic analysis (all results point to *G. lingzhi* in Clade A). However, because of the nomenclatural issue associated with many Genbank samples, it appears that our results change from *G. lucidum* to *G. lingzhi* (Table 1). This points to the importance of rectifying the Genbank taxonomy to avoid future, honest misidentifications.

## Supporting information

**S1 Fig. Bayesian phylogenetic analysis.** (A) Bayesian cladogram with Genbank accession numbers preceding species names for the reference panel. Samples are identified with reference to Table 1. Posterior probabilities greater than 0.50 are indicated along the branches. Branches with less than 0.50 posterior probability have been collapsed. Clade names A and B are from Loyd et al. [22] and Zhou et al. [24]. The red rectangle identifies the true *G. lucidum* samples per Loyd et al. [14] and the green rectangle contains the samples referred to as the *G. lingzhi* clade (many *G. lucidum* sequences are misidentified *G. lingzhi*). (B) Bayesian phylogram with unlabeled tips in the same order depicting branchlengths proportional to substitutions per site.
(TIFF)

**S1 Table. Three ITS primer pairs tested for amplification from reishi herbal supplements.**
(DOCX)

**S2 Table. Genbank reference panel sampling.**
(DOCX)

**S3 Table. Duplicate sequences removed from reference panel.** Identical sequences from the same species were removed to compress the alignment and facilitate phylogenetic analysis.
(DOCX)

**S4 Table. DNA concentration and purity for herbal supplement powder samples and fresh samples.**
(DOCX)

## Acknowledgments

The authors are grateful to the lab support staff in the Department of Biology that provided essential services throughout this study. Oregon Wild Harvest (Redmond, OR) supported TG throughout the duration of the study. Jonathan Eisen (UC Davis) kindly helped with references to the BLAST vs. phylogenetic analysis discussion.

## Author Contributions

**Conceptualization:** Tess Gunnels, Matthew Creswell, Janis McFerrin, Justen B. Whittall.

**Data curation:** Tess Gunnels, Justen B. Whittall.

**Formal analysis:** Tess Gunnels, Justen B. Whittall.

**Funding acquisition:** Tess Gunnels.

**Investigation:** Tess Gunnels, Justen B. Whittall.

**Methodology:** Tess Gunnels, Justen B. Whittall.

**Project administration:** Justen B. Whittall.

**Resources:** Justen B. Whittall.

**Software:** Justen B. Whittall.

**Supervision:** Justen B. Whittall.

**Validation:** Tess Gunnels, Justen B. Whittall.

**Visualization:** Justen B. Whittall.

**Writing – original draft:** Tess Gunnels, Justen B. Whittall.

**Writing – review & editing:** Tess Gunnels, Matthew Creswell, Janis McFerrin, Justen B. Whittall.

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
