## [Decision Letter · Decision Letter 0]

3 Sep 2020

PONE-D-20-21583

The ITS region provides a reliable DNA barcode for identifying reishi/lingzhi (Ganoderma) from herbal supplements

PLOS ONE

Dear Dr. Whittall,

Thank you for submitting your manuscript to PLOS ONE. After careful consideration, we feel that it has merit but does not fully meet PLOS ONE’s publication criteria as it currently stands. Therefore, we invite you to submit a revised version of the manuscript that addresses the points raised during the review process.

We look forward to receiving your revised manuscript.

Kind regards,

Tzen-Yuh Chiang

Academic Editor

PLOS ONE

Journal Requirements:

2. In addition, please ensure that you have deposited the sequencing data and provide the Genbank Accession numbers in Table 1 which are currently listed as XXXXX

"TG received funding from Oregon's Wild Harvest, an herbal supplement company included in our sampling. MC and JM are employees of Oregon's Wild Harvest. The senior author, JW received no funds from Oregon's Wild Harvest, nor any other herbal supplement company."

We note that one or more of the authors have an affiliation to the commercial funders of this research study : Oregon's Wild Harvest.

3.1. Please provide an amended Funding Statement declaring this commercial affiliation, as well as a statement regarding the Role of Funders in your study. If the funding organization did not play a role in the study design, data collection and analysis, decision to publish, or preparation of the manuscript and only provided financial support in the form of authors' salaries and/or research materials, please review your statements relating to the author contributions, and ensure you have specifically and accurately indicated the role(s) that these authors had in your study. You can update author roles in the Author Contributions section of the online submission form.

3.2. Please also provide an updated Competing Interests Statement declaring this commercial affiliation along with any other relevant declarations relating to employment, consultancy, patents, products in development, or marketed products, etc.  

Reviewers' comments:

Reviewer's Responses to Questions

**Comments to the Author**

1. Is the manuscript technically sound, and do the data support the conclusions?

Reviewer #1: No

Reviewer #2: Yes

2. Has the statistical analysis been performed appropriately and rigorously? 

Reviewer #1: Yes

Reviewer #2: Yes

3. Have the authors made all data underlying the findings in their manuscript fully available?

Reviewer #1: Yes

Reviewer #2: Yes

4. Is the manuscript presented in an intelligible fashion and written in standard English?

Reviewer #1: Yes

Reviewer #2: Yes

5. Review Comments to the Author

Reviewer #1: The manuscript PONE-D-20-21583 ‘The ITS region provides a reliable DNA barcode for identifying reishi/lingzhi (Ganoderma) from herbal supplements’ presents the results of sequencing ITS rDNA from commercial samples of powdered Ganoderma and several fresh control specimens, and compares two different downstream analyses to identify the sequences obtained: distance-based BLAST searches in public databases, and likelihood-based phylogenetic analyses. Unfortunately, I think that the manuscript does not provide enough scientific or technical novelties to be published in PLOS One. ITS rDNA is already known to discriminate most species of Ganoderma, and it has been already tested to identify powdered commercial samples. The most relevant result obtained is that the commercial samples analyzed are all mislabeled, suggesting that many marketed products need to be checked. This is important from a commercial point of view, but does not constitute a relevant scientific or technical novelty.

Therefore, I recommend the editors to reject the manuscript. I suggest the authors to look for a different journal to publish this work, or else explore other issues that could provide more interesting results in order to submit it again to PLOS One:

1) It would be great if the authors could quantify the different DNAs present in the samples by means of NGS, in order to estimate the amount of target fungus, additives and contaminants.

2) Another interesting issue is the great amount of misidentifications in public databases (what about UNITE?), and the need for reference sequences (preferably obtained from type collections) to stabilize the taxonomy of these species and provide more accurate database-guided identifications.

3) Authors could focus also on the comparison between distance-based and likelihood-based phylogenetic methods, and discuss cases where both methods produce different results (maybe comparing Ganoderma with other marketed species, i.e. Morchella?).

Below, I provide some more comments about the text:

Keywords = I would shorten the list of keywords. Please remove the words already included in the title. Please replace ‘nuclear ribosomal internal transcribed spacer region’ for ‘ITS nrDNA’. Correct ‘reishi’ (or remove it, as it is present in the title)

Abstract = The first 10 lines could be moved to the Introduction, or substantially reduced to one or two sentences. Please use cursive for species names.

‘clearly identified the predominant fungal DNA was G. lingzhi’ = the PCR product is supposed to belong to the predominant DNA present in the sample, but this could not be the case, as genus-specific primers were employed. Other species could be present and not amplified because of the primer specificity. You can change the sentence to ‘clearly identified the presence of G. lingzhi DNA’. Did you obtain a mixed signal after sequencing the PCRs done with universal primers?

‘ITS is a successful and cost-effective method for DNA-based species authentication’ = ITS is not a method, but a region of the rDNA. ‘ITS-based phylogenetic analysis is a successful and cost-effective method for DNA-based species authentication’ would be more correct.

Introduction

‘Barcoding is an efficient molecular tool’  better ‘Molecular barcoding is an efficient tool’

‘identifying morphologically, anatomically and biochemically enigmatic samples’ = Anatomy is a specific type of morphological study. Maybe you mean ‘macroscopically, microscopically and biochemically enigmatic samples’? Please correct other similar sentences.

‘especially helpful for maintaining the validity of active ingredients’ = DNA barcoding can detect the presence/absence of some target taxa, but it does not provide any info about the activity of ingredients. DNA of a species can be present after losing the biological activity of its compounds (i.e. after an aggresive processing).

‘avoiding contaminants that may cause allergic reactions for consumers’ = other species can be present also as additives. DNA testing can be employed also to check for their presence.

‘Reishi’ = please add a latin binomial with authors when this species is mentioned for the first time, or at least Ganoderma spp. if the vulgar name is applied to more than one species. Please add the authors of all scientific names when they are first mentioned.

‘The G. lucidum clade consists of several species that are in taxonomic flux’ = maybe better ‘The G. lucidum clade includes several phylogenetic lineages that received an unstable taxonomic treatment in the past’.

‘Ganoderma lucidum sensu lato (including G. ‘tsugae’)’ = Ganoderma tsugae is a valid name, so it does not need quotation marks. You can correct it as ‘(including G. tsugae Murrill)’.

‘can be found in the wild from Europe to northeastern China (likely escaped from cultivation in California and Utah, see [22])’ = there are three lineages within the G. lucidum clade (G. lucidum s. str., G. tsugae and G. oregonensis). Do you mean that G. tsugae and G. oregonensis could have escaped from cultivation? Or maybe you refer to Ganoderma luciudm s. str. instead?

‘According to several recent molecular phylogenetic studies, the taxonomy of G. lucidum and G. lingzhi remains uncertain [18, 26, 31].’ = I disagree, both species can be easily discriminated genetically. The whole paragraph needs to be corrected.

‘The nuclear ribosomal ITS region is a powerful tool for barcoding’ = better ‘the nuclear ribosomal internal transcribed spacer region (ITS) is an informative DNA region for barcoding...’

‘lineage-specific primers have been developed for many groups of fungi in order to improve PCR success’ = in many cases (especially for rDNA) the lineage-specific primers were not designed to improve PCR success, but to provide diagnostic primers to check the presence/absence of some species, or else to avoid the amplification of contaminant organisms.

‘Lineage-specific primers improve PCR success especially when working with compromised DNA templates that may be degraded, contain inhibitors, or be composed of a mixture of species’ = If DNA is degraded, specific primers by themselves will not improve PCR success. Maybe this could happen as a consequence of a different PCR approach, a smaller amplicon, or other factors unrelated with primer specificity. However, lineage-specific primers can improve PCR success when designed for non-conserved annealing regions if the ‘universal’ primers available do not work. In case that inhibitors are present, specific primers will not provide any improvement to PCR success. If a mixture of species is present, then specific primers can bypass the contaminants, improving PCR specificity (but not PCR success, which was also successful with the universal primers).

‘The two main approaches for analysis of the DNA sequences arising from barcoding investigations are similarity-based measures (e.g., best-hit BLAST or nearest neighbor analysis) and phylogenetic methods (e.g., maximum likelihood or Bayesian tree-building algorithms).’ = ‘distance’ is usually employed instead of ‘similarity’. You probably refer to genetic distance, but other types of distances can be measured, so please specify ‘genetic distance’. In addition, distance-based methods are employed also to build phylogenies, so they are phylogenetic methods. You could maybe call the second kind ‘likelihood-based methods’, since ML and bayesian approaches make use of the likelihood function.

‘Herein, we present an efficient barcoding method for unambiguous identification of the herbal supplement, reishi (G. lingzhi).’ = barcoding refers to the sequencing itself, but the identification needs also some kind of analysis, so maybe better ‘‘Herein, we present an efficient method for unambiguous identification of the herbal supplement, reishi (G. lingzhi).’

‘All powdered samples are morphologically unidentifiable as a mushroom’ = did you check for the presence of spores?

Table 1

I think it would be better to hide the name of stores and suppliers, unless you have the explicit consent of these companies to publish the results of your study. The species name is the one provided by the seller, or the results obtained from your analyses? Please clarify.

DNA extraction

‘Each nutritional supplement was extracted twice’ = you mean that two subsamples were taken from each sample and DNA was extracted from each of them, right? Please clarify.

‘Ganoderma-specific primers (G-ITSF1 and G-ITS-R2) were selected based on consistently producing strong single bands’ = you mean that the other primers produced multiple bands or weak bands?

‘These primers were designed to prevent amplification from plant or other fungal DNA, which is a common problem with herbal supplements since they often include a plant-based growing medium.’ = what about the fungal-specific primer ITS1F and the basidiomycete-specific primers ITS4B? Is contamination with other fungi a real issue in commercial samples of powdered Ganoderma?

‘ethidium bromide’ = I strongly recommend you to replace ethidium bromide with GelRed or other less toxic and contaminating DNA stain.

‘Forward and reverse chromatograms for each sample were trimmed to remove primer sequence and low quality sequence’ = chromatograms are trimmed to remove low quality reads. Primer sequences are rarely reached by the chromatogram, and they are almost always poorly resolved. However, they can be recovered in some cases. Also, you should correct ambiguous reads due to noise, dye blobs, and heteromorphic sites. So the most correct would be to say ‘Forward and reverse chromatograms for each sample were trimmed to remove low quality reads at the extremes, and edited to correct ambiguous reads and heteromorphic sites between them.’

BLAST = from which platform did you launch BLAST algorithm? Please cite Cochrane et al. (2011) if accessed from INSDC.

‘We also added all the unique Reishi samples (G. lingzhi and G. lucidum) of ITS using ENTREZ’ = what do you mean by ‘all the unique Reishi samples’? I dont understand, please clarify.

‘Finally, we included representatives of as many Ganoderma species we could find using a filtered discontiguous megablast’ = Maybe this is not the case, but there could be sequences related to your samples that are not listed in the BLAST results, especially if some species are overrepresented in GenBank. Maybe you could have just included those species more closely related to your samples by checking the phylogenetic studies available.

‘Several outgroup sequences were chosen’ = the outgroup should be selected from the clade most closely related to the sequences to be analyzed. In your case, this could be another species of Ganoderma, or the type of a sister genus.

‘Bayesian analysis used 1,500,000 Markov chains’ = usually 4-6 chains are employed. You probably mean 1.5 M generations.

‘and after a burn-in length of 750,000 samples.’ = 1.5 M generations sampled each 750 generations make a total 2000 sampled trees. So, you cannot remove 750.000 samples. You probably mean that you removed the samples taken during the first 750.000 generations (1000 samples, a 50% burn-in).

‘All primer pairs produced visible bands of expected size for the ITS region for both fresh and powdered samples. The Ganoderma-specific primers were chosen for all other analyses.’ = you should explain why these primers are chosen instead of the universal ones. Did you find problems in the sequences produced by the universal primers? Were they chosen to avoid putative contaminants? Please remove cursive from ITS.

‘Nucleotide sequences were recovered from the ITS region from all of the samples.’ = this sentence is superfluous. You should remove it and reorganize the paragraph.

‘After trimming the sequences, lengths ranged from 780 base pairs to 895 base pairs with an average of 854 base pairs.’ = you probably refer to the sequences obtained from GenBank, but it seems like you speak about the sequences produced de novo. Please clarify.

‘31 branches (36%) were greater than 70%’ = 70% is the bootstrap support, please specify it.

Fig. 1 = it would be better to show a phylogram instead of a cladogram. Also, you could add bayesian PP support to the nodes.

‘Fresh #3 had a sister relationship (to “G. lucidum” KX589244)’ = it is better so say ‘a significant relationship’. Change also in the following sentence. These relationships are based on very few bases from a single marker, so they should be interpreted cautiously.

‘Ganoderma brownii and falls clearly outside the G. lingzhi clade in a poorly resolved cluster of Ganoderma accessions in Clade A’ = why none of the two G. brownii ITS sequences in GenBank appear in the tree? Probably due to the sampling procedure. You could have ordered BLAST results by % similarity (removing those <50% coverage).

‘phylogenetically aberrant Genbank accessions’ = maybe better ‘putatively incorrectly named GenBank accessions’

‘a G. lucidum sample (MG654066) falls within a small, yet moderately supported clade of mostly North American samples’ = but in Fig. 1 you report that ‘The red rectangle [MG654066] identifies the only true G. lucidum sample per Loyd et al. [14]’ So, this is not ‘a phylogenetically aberrant Genbank accession’

‘Our study demonstrates that the ITS region provides an efficient barcode for store-bought

reishi herbal supplements as previously described by Loyd et al. [14] and Raja et al. [38].’ = maybe better to say that your study supports the conclusions reached by earlier authors.

‘Clade A sensu Zhou et al. [24] and Loyd et al. [14] which only includes “G. lucidum” as defined in the broadest sense’ = this is very confusing. Clade A sensu Loyd et al. includes multiple species, but not G. lucidum. You could say that clade A includes species morphologically similar to G. lucidum. A sensu lato always includes the sensu stricto plus other clades.

‘Technically all of our samples are misidentified since they are being sold as “G. lucidum”, yet are molecularly allied with the G. lingzhi samples in Clade A.’ = So, MG654066 is not an aberrant accesion, but the correct concept of G. lucidum.

‘Genbank sample MG654066 named G. lucidum (in Clade B) is the only sample that represents G. lucidum sensu stricto’ = this seems to mean that MG654066 is the only known sequence of G. lucidum s. str. However, there are others (at least 8 more in Loyd et al.). It is the only one in your tree, probably because of the sampling process employed.

‘there were distinct phylogenetic affinities clearly indicating separate sources’ = this could be true, but not necessarily.

‘Our results were generally robust to whether we used the entire ITS region or the trimmed region’ = not for G. brownii. This should be enough to say that the sampling method employed to obtain closely related sequences from GenBank was not the most suitable one. I think you should have ordered GenBank results by their similarity with the query, not the BLAST score.

Reviewer #2: Comments

The authors employed a molecular phylogenetic analysis for species authentication of the commonly used fungal supplement, reishi (Ganoderma lingzhi), by amplifying and sequencing the nuclear ribosomal internal transcribed spacer regions (ITS) with genus-specific primers.Their investigation indicated that ITS region could be used in the herbal supplement industry for fungal and plant species that are difficult to identify. This research is of general interest. There are some major comments for authors’ revision.

1. Could the authors please simplify the introduction? Eg. [reviewed in 15, 16], (yet see [18, 19] for biochemical profiles of reishi and close relatives) could be shorted as citations (just keep the reference number).

2. Could the authors please show the features of all the samples? It will be more intuitive for the readers to know the difference of the samples.

3. Please fill in the GenBank Accession numbers in Table 3 (the third column).

4. Could the authors please show the inter/intra-specific distance among these samples?

5. In view of the primers used in this study was Ganoderma-specific ones, how could the authors determine if there is any adulteration derived from other genera in the commercial samples?

6. The discussion section could be separated into several parts according to a clear logic.

6. PLOS authors have the option to publish the peer review history of their article (what does this mean?). If published, this will include your full peer review and any attached files.

Reviewer #1: No

Reviewer #2: No

---

## [Author Response · Author response to Decision Letter 0]

19 Oct 2020

Response to Reviewer’s Comments

Gunnel’s et al. Ganoderma barcoding

PONE-D-20-21583

Dear Dr. Tzen-Yuh Chiang, Academic Editor, PLOS ONE

We were glad to see your email indicating that you think our paper “has merit”. We are thankful for the two reviews, especially Reviewer #1’s thorough comments and valiant efforts in helping us improve our manuscript. We know this revised version is much better for their careful attention and thoughtful suggestions. A question for you:

1. Reviewer #1 wrote in regard to Table 1, “I think it would be better to hide the name of stores and suppliers, unless you have the explicit consent of these companies to publish the results of your study.” Please advise.

We have addressed all of the “Additional Journal Requirements” included in your email including amended Statements regarding Funding and Competing Interests that you can change in the online submission on my behalf. 

#1 – checked formatting and reformatted as necessary

#2 – Genbank Accession numbers now in Table 2

#3.1 Amended Funding Statement: “There is a commercial affiliation of some authors (MC and JM) and the herbal supplement industry. Oregon’s Wild Harvest provided funding for supplies, sequencing and hourly wages to an undergraduate researcher (TG). MC and JM contributed to the study design and preparation of the manuscript, but in a completely unbiased way. They just wanted to know if their product was correctly identified or not.”

#3.2 Amended Competing Interests Statement: "MC and JM are employees of Oregon's Wild Harvest, an herbal supplement company included in our sampling. As an undergraduate researcher, TG was partially funded by Oregon’s Wild Harvest. This does not alter our adherence to PLOS ONE policies on sharing data and materials. There are no other relevant declarations relating to employment, consultancy, patents, products in development, or marketed products, etc.”

Below we have summarized both reviewers’ comments in italics followed by our indented responses quoting from the revised manuscript whenever we felt it would be helpful in interpreting our revision.

Reviewer #1

General Comment: Unfortunately, I think that the manuscript does not provide enough scientific or technical novelties to be published in PLOS One.

Our understanding of the seven criteria for publication at PLOS ONE does not include anything about “scientific or technical novelties”.

1) It would be great if the authors could quantify the different DNAs present in the samples by means of NGS, in order to estimate the amount of target fungus, additives and contaminants.

We are currently pursuing this as a follow-up study, but the COVID19 response has prevented us from entering the lab since February, 2020. Therefore, we chose to report our Sanger-based results first, then if/when the pandemic-response allows, we will get back to trying NGS approaches on these samples. We feel our results are robust and a worthy contribution to the scientific literature.

2) Another interesting issue is the great amount of misidentifications in public databases (what about UNITE?), and the need for reference sequences (preferably obtained from type collections) to stabilize the taxonomy of these species and provide more accurate database-guided identifications.

Thank you for pointing out this very interesting database - UNITE. We found numerous records for G. lucidum and were initially hopeful to gain some valuable insights from its integration with “species hypotheses” based on 0.5-3% ITS sequene divergence. Unfortunately, the search tool retrieved zero records for G. lingzhi. Furthermore, I defer to Ganoderma taxonomic experts in circumscribing species boundaries that should include geographic, morphological and biochemical data when available to make inferences about reproductive isolation and species status (three forms of data that UNITE does not include in its results). Therefore, I don’t think using the database will contribute anything to our study since our study focuses on trying to differentiate these two taxa, but one is not in the database. We have added a reference to this tool in our Discussion of barcoding with the ITS region so readers will be more likely to attempt to utilize it (hopefully with more success than we had) (Nilsson et al. Nucleic Acids Research 2018).

We agree that reference sequences from type collections would be helpful, but we don’t have access to these and so their sampling is beyond the scope of our current project.

3) Authors could focus also on the comparison between distance-based and likelihood-based phylogenetic methods, and discuss cases where both methods produce different results (maybe comparing Ganoderma with other marketed species, i.e. Morchella?).

Although we feel that the comparison of using genetic distance-based approaches to barcoding identifications vs. phylogenetic ones are important discussions, since our results showed very little differences (qualitatively at least), we don’t think it’s appropriate to dive deep into this topic in this manuscript. We also cite some foundational papers that identified the circumstances in which the two will disagree in our penultimate paragraph of the Introduction (see refs 43-46). 

Regarding the suggestion to pursue a comparison with morels, thank you for pointing us to this very interesting example. We have integrated Du et al.’s 2012 study “How well do ITS rDNA sequences differentiate species of true morels (Morchella)?” into our Discussion on using ITS for fungal species identifications. Interestingly, their webtool produces distance-based trees (UPGMA and Neighbor Joining), yet they report parsimony trees in their manuscript. We have left the distance vs. likelihood debate to be resolved in the phylogenetics literature as we are not qualified to address that here.

More comments about the text from Reviewer #1:

Keywords = I would shorten the list of keywords. Please remove the words already included in the title. Please replace ‘nuclear ribosomal internal transcribed spacer region’ for ‘ITS nrDNA’. Correct ‘reishi’ (or remove it, as it is present in the title)

PLOS ONE General Information for Authors states, “Add keywords to help expedite processing of your manuscript (optional). You will not have an opportunity to make changes, so make sure to add concise, accurate keywords now.” There is no additional guidance on the number or redundancy with the title in this regard. We have removed any terms that are redundant with the title. We kept nuclear ribosomal internal transcribed spacer region since it was abbreviated in the title.

Abstract

The first 10 lines could be moved to the Introduction, or substantially reduced to one or two sentences. 

We removed several sentences at the beginning of the abstract in response to this comment.

Please use cursive for species names.

We have confirmed that all Latin names are italicized. I think some formatting is lost in the abstract during the submission process, but it is correct in the downloadable WORD version of the manuscript.

‘clearly identified the predominant fungal DNA was G. lingzhi’ = the PCR product is supposed to belong to the predominant DNA present in the sample, but this could not be the case, as genus-specific primers were employed. Other species could be present and not amplified because of the primer specificity. You can change the sentence to ‘clearly identified the presence of G. lingzhi DNA’. Did you obtain a mixed signal after sequencing the PCRs done with universal primers?

Good point. We changed the text as suggested. We only sequenced PCR products from the Ganoderma-specific primers, but all primer pairs tested (including universal primers) produced single strong bands.

‘ITS is a successful and cost-effective method for DNA-based species authentication’ = ITS is not a method, but a region of the rDNA. ‘ITS-based phylogenetic analysis is a successful and cost-effective method for DNA-based species authentication’ would be more correct.

 Fixed.

Introduction

‘Barcoding is an efficient molecular tool’  better ‘Molecular barcoding is an efficient tool’

Done.

‘identifying morphologically, anatomically and biochemically enigmatic samples’ = Anatomy is a specific type of morphological study. Maybe you mean ‘macroscopically, microscopically and biochemically enigmatic samples’? Please correct other similar sentences.

Done.

‘especially helpful for maintaining the validity of active ingredients’ = DNA barcoding can detect the presence/absence of some target taxa, but it does not provide any info about the activity of ingredients. DNA of a species can be present after losing the biological activity of its compounds (i.e. after an aggressive processing). ‘avoiding contaminants that may cause allergic reactions for consumers’ = other species can be present also as additives. DNA testing can be employed also to check for their presence.

Done. Now reads, “…especially helpful for authenticating the ingredients and avoiding contaminants and additives that may cause allergic reactions for consumers [8], although DNA of a species can be present long after losing the biological activity of its compounds (i.e. after an aggressive processing).”

‘Reishi’ = please add a latin binomial with authors when this species is mentioned for the first time, or at least Ganoderma spp. if the vulgar name is applied to more than one species. Please add the authors of all scientific names when they are first mentioned.

 Done.

‘The G. lucidum clade consists of several species that are in taxonomic flux’ = maybe better ‘The G. lucidum clade includes several phylogenetic lineages that received an unstable taxonomic treatment in the past’.

 Done

‘Ganoderma lucidum sensu lato (including G. ‘tsugae’)’ = Ganoderma tsugae is a valid name, so it does not need quotation marks. You can correct it as ‘(including G. tsugae Murrill)’.

 Done

‘can be found in the wild from Europe to northeastern China (likely escaped from cultivation in California and Utah, see [22])’ = there are three lineages within the G. lucidum clade (G. lucidum s. str., G. tsugae and G. oregonensis). Do you mean that G. tsugae and G. oregonensis could have escaped from cultivation? Or maybe you refer to Ganoderma luciudm s. str. instead?

 Unsure, so changed to “…some have likely escaped…” to be conservative.

‘According to several recent molecular phylogenetic studies, the taxonomy of G. lucidum and G. lingzhi remains uncertain [18, 26, 31].’ = I disagree, both species can be easily discriminated genetically. The whole paragraph needs to be corrected. 

Thank you for clarifying. We’ve restructured the last two sentences of the paragraph that now read, “The history of taxonomic confusion surrounding G. lucidum and G. lingzhi [18, 26, 31] has been largely resolved by recent work that clearly identifies two distinct lineages based on morphology [23], biochemistry [18], and molecular phylogenetics [22, 23, 24, 32].” The first part about Ganoderma in general and introducing the two species at hand is not invalidated by the reviewer’s comment.

‘The nuclear ribosomal ITS region is a powerful tool for barcoding’ = better ‘the nuclear ribosomal internal transcribed spacer region (ITS) is an informative DNA region for barcoding...’

 Done.

‘lineage-specific primers have been developed for many groups of fungi in order to improve PCR success’ = in many cases (especially for rDNA) the lineage-specific primers were not designed to improve PCR success, but to provide diagnostic primers to check the presence/absence of some species, or else to avoid the amplification of contaminant organisms. ‘Lineage-specific primers improve PCR success especially when working with compromised DNA templates that may be degraded, contain inhibitors, or be composed of a mixture of species’ = If DNA is degraded, specific primers by themselves will not improve PCR success. Maybe this could happen as a consequence of a different PCR approach, a smaller amplicon, or other factors unrelated with primer specificity. However, lineage-specific primers can improve PCR success when designed for non-conserved annealing regions if the ‘universal’ primers available do not work. In case that inhibitors are present, specific primers will not provide any improvement to PCR success. If a mixture of species is present, then specific primers can bypass the contaminants, improving PCR specificity (but not PCR success, which was also successful with the universal primers).

Corrected. These sentences now read, “However, lineage-specific primers have been developed for many groups of fungi to help diagnose the presence or absence of particular species [23]. Lineage-specific primers can also improve PCR specificity especially when working with compromised DNA templates…”

 ‘The two main approaches for analysis of the DNA sequences arising from barcoding investigations are similarity-based measures (e.g., best-hit BLAST or nearest neighbor analysis) and phylogenetic methods (e.g., maximum likelihood or Bayesian tree-building algorithms).’ = ‘distance’ is usually employed instead of ‘similarity’. You probably refer to genetic distance, but other types of distances can be measured, so please specify ‘genetic distance’. In addition, distance-based methods are employed also to build phylogenies, so they are phylogenetic methods. You could maybe call the second kind ‘likelihood-based methods’, since ML and bayesian approaches make use of the likelihood function.

 Changed “similarity-based” to “genetic distance-based” here and throughout.

‘Herein, we present an efficient barcoding method for unambiguous identification of the herbal supplement, reishi (G. lingzhi).’ = barcoding refers to the sequencing itself, but the identification needs also some kind of analysis, so maybe better ‘‘Herein, we present an efficient method for unambiguous identification of the herbal supplement, reishi (G. lingzhi).’

 Fixed.

‘All powdered samples are morphologically unidentifiable as a mushroom’ = did you check for the presence of spores?

We were able to check one of the powdered samples and only saw mycelium. No fruiting bodies = no spores. We changed this section of the Methods to: “All powdered samples were macroscopically unidentifiable as a mushroom and for Powder #1, only mycelium was identifiable under compound microscope (40-100x) (Creswell and McFerrin, unpublished data).”

Table 1

I think it would be better to hide the name of stores and suppliers, unless you have the explicit consent of these companies to publish the results of your study. The species name is the one provided by the seller, or the results obtained from your analyses? Please clarify. 

We emailed the PLOS ONE handling editor (Dr. Chiang) on Sept. 5, 2020 regarding the naming of retailers and suppliers in our manuscript. We are still waiting to hear from him so have not removed it yet. In the meantime, we have clarified that the Species names listed in the table are “as advertised”.

DNA extraction

‘Each nutritional supplement was extracted twice’ = you mean that two subsamples were taken from each sample and DNA was extracted from each of them, right? Please clarify.

 Yes, clarified.

‘Ganoderma-specific primers (G-ITSF1 and G-ITS-R2) were selected based on consistently producing strong single bands’ = you mean that the other primers produced multiple bands or weak bands?

We mean both. Therefore, our explanation about “consistently producing strong, single bands” shouldn’t be misinterpreted. We added a comma to clarify these two characteristics.

‘These primers were designed to prevent amplification from plant or other fungal DNA, which is a common problem with herbal supplements since they often include a plant-based growing medium.’ = what about the fungal-specific primer ITS1F and the basidiomycete-specific primers ITS4B? Is contamination with other fungi a real issue in commercial samples of powdered Ganoderma?

Our statement refers specifically to the Ganoderma-specific primers (G-ITS-F1 and G-ITS-R2) as stated in the prior sentence. We list the fungal specific primers mentioned in the Reviewer’s comment in Table 1 as having tried them, but chose the Ganoderma-specific primers for reasons mentioned above. 

Regarding contamination with other fungi in powdered Ganoderma, Loyd found only close relatives (e.g. G. applanatum). As thoughtfully mentioned, our choice of Ganoderma-specific primers would reduce the likelihood of detecting non-Ganoderma contamination. However, (1) previous studies have shown if there is any adulteration it is within the polypores and (2) we prove that these primers can amplify other distantly related Ganoderma species (Fresh #1 = G. brownii) and even species from another polypore genus (Fresh #2 = Fomitopsis pinicola). Therefore, I think we are safe using these primers to test for accidentally or intentionally substituted ingredients in reishi herbal supplements.

‘ethidium bromide’ = I strongly recommend you to replace ethidium bromide with GelRed or other less toxic and contaminating DNA stain.

Thanks for the suggestion. I didn’t know ethidium bromide was susceptible to contamination. We are planning to switch as soon as our stocks of EtBr run out.

‘Forward and reverse chromatograms for each sample were trimmed to remove primer sequence and low-quality sequence’ = chromatograms are trimmed to remove low quality reads. Primer sequences are rarely reached by the chromatogram, and they are almost always poorly resolved. However, they can be recovered in some cases. Also, you should correct ambiguous reads due to noise, dye blobs, and heteromorphic sites. So the most correct would be to say ‘Forward and reverse chromatograms for each sample were trimmed to remove low quality reads at the extremes, and edited to correct ambiguous reads and heteromorphic sites between them.’

Thank you for trying to clarify this section. However, the ends are not “reads”, but portions of a single contiguous sequence. We changed this to “…trimmed to remove the opposing primer sequence and low-quality sequence at the beginning and end of each read and edited to correct any ambiguous base calls.” We did not have any heteromorphic sites (and this term has multiple meanings so we want to avoid confusion). Furthermore, we definitely sequenced the opposing primer sequence and had to remove it so we clarified that this is the primer that lands on the opposing side of the direction of sequencing.

BLAST = from which platform did you launch BLAST algorithm? Please cite Cochrane et al. (2011) if accessed from INSDC.

We used BLAST directly through NCBI’s portal (no secondary references required). Therefore, the original references (51-53 listed below) seem most relevant:

51. Altschul SF, Gish W, Miller W, Myers EW, Lipman DJ. Basic local alignment search tool. J Mol Biol. 1990 Oct 5;215(3):403-10.

52. Zhang Z, Schwartz S, Wagner L, Miller W. A greedy algorithm for aligning DNA sequences. J Comput Biol. 2000 Feb 1;7(1-2):203-14.

53. Morgulis A, Coulouris G, Raytselis Y, Madden TL, Agarwala R, Schäffer AA. Database indexing for production MegaBLAST searches. Bioinformatics. 2008 Aug 15;24(16):1757-64.

‘We also added all the unique Reishi samples (G. lingzhi and G. lucidum) of ITS using ENTREZ’ = what do you mean by ‘all the unique Reishi samples’? I dont understand, please clarify.

ENTREZ is a text search portal to NCBI’s Genbank. It’s a good way to get started finding sequences, but BLAST is an essential follow-up to ensure you have all similar sequences even if the text doesn’t match (e.g. misidentified sequences). We clarified the use of ENTREZ by writing, “…using text searches in ENTREZ.”

‘Finally, we included representatives of as many Ganoderma species we could find using a filtered discontiguous megablast’ = Maybe this is not the case, but there could be sequences related to your samples that are not listed in the BLAST results, especially if some species are overrepresented in GenBank. Maybe you could have just included those species more closely related to your samples by checking the phylogenetic studies available.

If the sequences were “related” to our sample, BLAST would have found them. I don’t know of any errors in the search algorithm regarding “overrepresentation”. As long as one expands the number of reported sequences so it doesn’t max out, they will get everything that is similar given the search parameters. Furthermore, we checked some key phylogenetic studies to ensure we had all the relevant sequences in our reference panel. 

‘Several outgroup sequences were chosen’ = the outgroup should be selected from the clade most closely related to the sequences to be analyzed. In your case, this could be another species of Ganoderma, or the type of a sister genus.

Since we surveyed Ganoderma species broadly, we chose a more distant outgroup. Since we recovered the main three clades of Ganoderma that are previously described, it doesn’t appear to make a difference. I would be more concerned about accidentally choosing something too close and then the outgroup rooting would have severe phylogenetic repercussions. No changes made in this regard.

‘Bayesian analysis used 1,500,000 Markov chains’ = usually 4-6 chains are employed. You probably mean 1.5 M generations.

‘and after a burn-in length of 750,000 samples.’ = 1.5 M generations sampled each 750 generations make a total 2000 sampled trees. So, you cannot remove 750.000 samples. You probably mean that you removed the samples taken during the first 750.000 generations (1000 samples, a 50% burn-in).

Thank you for helping clarify our methods. In our reanalysis, we ran more generations. It now reads, “Bayesian analysis ran for 2,000,000 generations. After removing the first 1,000,000 generations as burn-in, we sampled trees every 1000 generations creating a posterior distribution of 1000 trees.”

Results

‘All primer pairs produced visible bands of expected size for the ITS region for both fresh and powdered samples. The Ganoderma-specific primers were chosen for all other analyses.’ = you should explain why these primers are chosen instead of the universal ones. Did you find problems in the sequences produced by the universal primers? Were they chosen to avoid putative contaminants? Please remove cursive from ITS.

We have removed the italics for ITS and clarified our primer selection in the following revised sentence, “The PCR products using Ganoderma-specific primers were chosen for sequencing and all subsequent analyses based on their increased band intensity compared to other primers.”

‘Nucleotide sequences were recovered from the ITS region from all of the samples.’ = this sentence is superfluous. You should remove it and reorganize the paragraph.

 Done.

‘After trimming the sequences, lengths ranged from 780 base pairs to 895 base pairs with an average of 854 base pairs.’ = you probably refer to the sequences obtained from GenBank, but it seems like you speak about the sequences produced de novo. Please clarify.

No, this statement refers to the seven new samples from herbal supplements. We have clarified in the following revised sentence, “After trimming these newly created sequences for seven samples…”

‘31 branches (36%) were greater than 70%’ = 70% is the bootstrap support, please specify it.

 Fixed.

Fig. 1 = it would be better to show a phylogram instead of a cladogram. Also, you could add bayesian PP support to the nodes.

Fig 1 has been completely redone since we included two G. brownii sequences and three additional G. lucidum sequences. We have included a phylogram as an inset (B) without labels (to save space) to depict the relative branchlengths as suggested. We also superimposed the Bayesian posterior probabilities onto the maximum likelihood tree at critical nodes.

‘Fresh #3 had a sister relationship (to “G. lucidum” KX589244)’ = it is better so say ‘a significant relationship’. Change also in the following sentence. These relationships are based on very few bases from a single marker, so they should be interpreted cautiously.

Completely rewritten.

‘Ganoderma brownii and falls clearly outside the G. lingzhi clade in a poorly resolved cluster of Ganoderma accessions in Clade A’ = why none of the two G. brownii ITS sequences in GenBank appear in the tree? Probably due to the sampling procedure. You could have ordered BLAST results by % similarity (removing those <50% coverage).

Thank you for noticing that oversight on our part. We have redone the phylogenetic analyses with these G. brownii sequences and the missing G. lucidum sequences mentioned below in regard to MG654066. They all fall out in the tree as expected, but the study is much more robust having updated this portion.

‘phylogenetically aberrant Genbank accessions’ = maybe better ‘putatively incorrectly named GenBank accessions’

Done (also at the end of the next paragraph describing the consistencies in the Bayesian analysis).

‘a G. lucidum sample (MG654066) falls within a small, yet moderately supported clade of mostly North American samples’ = but in Fig. 1 you report that ‘The red rectangle [MG654066] identifies the only true G. lucidum sample per Loyd et al. [14]’ So, this is not ‘a phylogenetically aberrant Genbank accession’

Correct. We have now added the additional G. lucidum sequences and removed any references to this single sample being “aberrant”.

Discussion

‘Our study demonstrates that the ITS region provides an efficient barcode for store-bought reishi herbal supplements as previously described by Loyd et al. [14] and Raja et al. [38].’ = maybe better to say that your study supports the conclusions reached by earlier authors.

 Done.

‘Clade A sensu Zhou et al. [24] and Loyd et al. [14] which only includes “G. lucidum” as defined in the broadest sense’ = this is very confusing. Clade A sensu Loyd et al. includes multiple species, but not G. lucidum. You could say that clade A includes species morphologically similar to G. lucidum. A sensu lato always includes the sensu stricto plus other clades.

We agree that this portion of the Discussion was poorly written. We have reduced it to the essential details in the following revised sentence, “All of the samples we examined had BLAST and phylogenetic results suggesting they were clearly members of Clade A sensu Zhou et al. [24] and Loyd et al. [14].”

‘Technically all of our samples are misidentified since they are being sold as “G. lucidum”, yet are molecularly allied with the G. lingzhi samples in Clade A.’ = So, MG654066 is not an aberrant accession, but the correct concept of G. lucidum.

 Correct. We have now added the additional G. lucidum sequences and removed any references to this single sample being “aberrant”.

‘Genbank sample MG654066 named G. lucidum (in Clade B) is the only sample that represents G. lucidum sensu stricto’ = this seems to mean that MG654066 is the only known sequence of G. lucidum s. str. However, there are others (at least 8 more in Loyd et al.). It is the only one in your tree, probably because of the sampling process employed.

 Correct. We have now added the additional G. lucidum sequences and removed any references to this single sample being “aberrant”. Since several sequences were identical, we added a row to S3 Table indicating which G. lucidum samples were removed as identical (see below for detailed explanation).

We searched Genbank for “Ganoderma lucidum[organism] AND Loyd[author]” and got 10 hits (including our original MG654066). A quick alignment showed that of these sequences there were only four unique sequences as you can see from the distance matrix appended below. Therefore, we added three additional sequences and indicated in Supplemental Table 3 that there are seven identical sequences of which we chose MH160071 as our representative. Our new tree has four G. lucidum sequences instead of the single placeholder in our original submission.

Table S3 now has the following row for the G. lucidum sequences included in the phylogenetic analysis (MH160071) and the five other sequences identical to it:

MH160071 MG654067, MG654070, MG654071, MG654072, MG654073

Our sampling table for the reference panel (S2 Table) has also been updated with the new G. lucidum sequences as well.

‘there were distinct phylogenetic affinities clearly indicating separate sources’ = this could be true, but not necessarily.

OK, we changed “clearly indicating” to “suggesting” leaving room for alternative interpretations yet emphasizing the most likely conclusion.

‘Our results were generally robust to whether we used the entire ITS region or the trimmed region’ = not for G. brownii. This should be enough to say that the sampling method employed to obtain closely related sequences from GenBank was not the most suitable one. I think you should have ordered GenBank results by their similarity with the query, not the BLAST score.

Yes, as Reviewer #1 pointed out above, G. brownii (and some G. lucidum sequences) were accidentally removed from the alignment. I don’t think this was an inherent flaw in the sorting method (Genbank reports BLAST results based on MaxScore which is a composite of the E-value, % identity and % coverage and is widely used to identify most likely matches in Genbank).

Instead of a methodological error, I believe the missing sequences are simply an oversight on our part. This happened when we removed identical sequences from the reference panel because they were stalling the phylogenetic analyses (and by definition provide no additional phylogenetic information). When these identical sequences were removed, they were to be placed in Supplemental Table for reference, but thankfully Reviewer #1 noticed that some were lost in the process. We have reincluded the missing G. brownii and G. lucidum sequences, triple checked the results for any other missing sequences (none found) and updated the alignment & phylogenetic analyses, produced new trees for Figure 1 and the Supplemental Figure (Bayesian tree), and rewritten the relevant sections of the Results and Discussion. Qualitatively, we didn’t change our major findings, but the reference panel is now complete and the results even more robust than before.

Reviewer #2

1. Could the authors please simplify the introduction? Eg. [reviewed in 15, 16], (yet see [18, 19] for biochemical profiles of reishi and close relatives) could be shorted as citations (just keep the reference number).

 Done.

2. Could the authors please show the features of all the samples? It will be more intuitive for the readers to know the difference of the samples.

We are awaiting feedback from the Associate Editor in regard to what to show and what to keep confidential. If the Associate Editor suggests we show the products, then we can try to assemble a multi-panel plate as suggested.

3. Please fill in the GenBank Accession numbers in Table 3 (the third column).

 Genbank Accession numbers have been added to the table.

4. Could the authors please show the inter/intra-specific distance among these samples?

We have integrated this into the Results section when discussing the genetic variation detected among reishi samples and the field collected controls.

5. In view of the primers used in this study was Ganoderma-specific ones, how could the authors determine if there is any adulteration derived from other genera in the commercial samples?

Great point. We assumed that any adulteration intentional or accidental would have been by substitution of G. lingzhi with another polypore (mostly likely in the same genus, but possibly in an allied genus like Fomitopsis) based on the findings of Loyd. Fortunately, the primers are optimized for polypores and we have confirmed this empirically by successful amplification and sequencing of the control samples that represent a distant relative within Ganoderma and another sample that amplified with these primers and was identified Fomitopsis. We have clarified this point in the Discussion.

6. The discussion section could be separated into several parts according to a clear logic.

According to PLOS ONE Guide to Authors, subheadings in the Discussion are optional. Since there are only six paragraphs, most subheadings would only apply to a single paragraph and be less of an organizational tool. They would break-up the flow of the Discussion that we were shooting for. No change made.

---

## [Decision Letter · Decision Letter 1]

28 Oct 2020

The ITS region provides a reliable DNA barcode for identifying reishi/lingzhi (Ganoderma) from herbal supplements

PONE-D-20-21583R1

Dear Dr. Whittall,

We’re pleased to inform you that your manuscript has been judged scientifically suitable for publication and will be formally accepted for publication once it meets all outstanding technical requirements.

Kind regards,

Tzen-Yuh Chiang

Academic Editor

PLOS ONE

Additional Editor Comments (optional):

Reviewers' comments:

Reviewer's Responses to Questions

**Comments to the Author**

1. If the authors have adequately addressed your comments raised in a previous round of review and you feel that this manuscript is now acceptable for publication, you may indicate that here to bypass the “Comments to the Author” section, enter your conflict of interest statement in the “Confidential to Editor” section, and submit your "Accept" recommendation.

Reviewer #2: All comments have been addressed

2. Is the manuscript technically sound, and do the data support the conclusions?

Reviewer #2: Yes

3. Has the statistical analysis been performed appropriately and rigorously? 

Reviewer #2: Yes

4. Have the authors made all data underlying the findings in their manuscript fully available?

Reviewer #2: Yes

5. Is the manuscript presented in an intelligible fashion and written in standard English?

Reviewer #2: Yes

6. Review Comments to the Author

Reviewer #2: (No Response)

7. PLOS authors have the option to publish the peer review history of their article (what does this mean?). If published, this will include your full peer review and any attached files.

Reviewer #2: No

---

## [Editor Report · Acceptance letter]

4 Nov 2020

PONE-D-20-21583R1 

The ITS region provides a reliable DNA barcode for identifying reishi/lingzhi (*Ganoderma*) from herbal supplements 

Dear Dr. Whittall:

I'm pleased to inform you that your manuscript has been deemed suitable for publication in PLOS ONE. Congratulations! Your manuscript is now with our production department. 

Kind regards, 

on behalf of

Dr. Tzen-Yuh Chiang 

Academic Editor

PLOS ONE